# Algorithmically Optimized Hemispherical Dome as a Secondary Optical Element for the Fresnel Lens Solar Concentrator

**Hassan Qandil [1], Shuping Wang [2] and Weihuan Zhao [1,\*]**

[1]  Mechanical and Energy Engineering Department, University of North Texas, Denton, TX 76207, USA
[2]  Engineering Technology Department, University of North Texas, Denton, TX 76207, USA
\*  Correspondence: weihuan.zhao@unt.edu; Tel.: +1-940-369-5929

**Featured Application: solar heating; concentrated photovoltaics.**

**Abstract:** The significance of this work lies in the development of a novel code-based, detailed, and deterministic geometrical approach that couples the optimization of the Fresnel lens primary optical element (POE) and the dome-shaped secondary optical element (SOE). The objective was to maximize the concentration acceptance product (CAP), while using the minimum SOE and receiver geometry at a given f-number and incidence angle (also referred to as the tracking error angle). The laws of polychromatic light refraction along with trigonometry and spherical geometry were utilized to optimize the POE grooves, SOE radius, receiver size, and SOE–receiver spacing. Two literature case studies were analyzed to verify this work's optimization, both with a spot Fresnel lens POE and a spherical dome SOE. Case 1 had a 625 cm$^2$ POE at an f-number of 1.5, and Case 2 had a 314.2 cm$^2$ POE at an f-number of 1.34. The equivalent POE designed by this work, with optimized SOE radiuses of 13.6 and 11.4 mm, respectively, enhanced the CAP value of Case 1 by 52% to 0.426 and that of Case 2 by 32.4% to 0.45. The SOE's analytical optimization of Case 1 was checked by a simulated comparative analysis to ensure the validity of the results. Fine-tuning this design for thermal applications and concentrated photovoltaics is also discussed in this paper. The algorithm can be further improved for more optimization parameters and other SOE shapes.

**Keywords:** solar concentrator; solar energy; renewable energy; optics; Fresnel lens; secondary optics; non-imaging optics

## 1. Introduction

Recent fluctuations in oil prices have raised enormous doubts about the energy security of many nations. These fluctuations have contributed to the need for low-cost and efficient harvesting of the sun's energy—a pursuit which has greatly increased the technological advancements in the design and fabrication of solar equipment [1]. Solar research equipment can achieve remarkably high temperatures, which can be used either directly or in a thermal cycle to generate electricity. Fresnel lens concentrators have emerged as promising alternatives to reflective mirrors, especially for their lower investment costs, competitive optical performance, and compact size [2].

The use of secondary optical elements (SOEs) has been analyzed in order to widen the acceptance angle of the Fresnel lens primary optical element (POE), improve the optical efficiency, and enhance the flux uniformity for concentrated photovoltaic (CPV) applications [3,4].

The majority of prior solar research has used geometrical optics and the edge-ray principle as the design method of choice for the different shapes of SOEs. For instance, a four-fold Fresnel–Kohler

concentrator was designed by the edge-ray principle in a four-part symmetry relating the POE and the SOE [5–7]. On the same note, the edge-ray principle has been used in the SOE designs of a ball-lens [8,9], a dome [10,11], a kaleidoscope with a flat top surface [12–18] and a domed top surface [12–14,18,19], a half-egg [12,13], a refractive truncated pyramid (RTP) [15–17], a single-lens optical element (SILO) pyramid [15–17,19], and a refractive dielectric-crossed compound parabolic concentrator (DCCPC) [15–17].

While a few prior studies have used iterative simulations to optimize the SOE geometry [20], the analytical design in much of the literature has not been typically explained in detailed equations, which makes it hard to replicate, enhance the SOE optimization, or include it in a simulative computer code.

Refractive SOEs have advantages over reflective ones, because they are less material-intensive and are easier to manufacture—not only from plastic but also from glass [11]. However, the receiver (which is usually a solar cell) is normally attached with an optical adhesive at the bottom of the refractive SOE [6,13,17,19]. This might jeopardize the optical performance and limit the long-term system reliability. Instead, introducing an air gap between the receiver and the SOE can reduce those risks [8].

It can also be noted that some solar literature does not account for a solar incidence angle in the design phases of the POE and/or SOE [8,9], while other work does not optimize the POE along with the SOE to further fine-tune the SOE geometry [8,10], and thus this does not enhance the optical performance. Another issue has been the lack of the polychromatic representation of light in the POE and/or SOE design phases [9,14], which can deviate the simulated analysis from the experimental results.

This work aims to enhance and detail the use of geometrical optics in an algorithmic approach to collectively design a solar concentrator with a POE and a single hemispherical SOE. The goal was to achieve a high concentration acceptance product (CAP) at a predetermined incidence (tracking error) angle and f-number, while minimizing the geometry of the SOE and the receiver, and introducing an air gap in-between them. The optimization employed polychromatic ray tracing, starting with a light incident at the upper POE surface until refracted through the bottom SOE surface. The resulting homogeneity of the focal irradiance could be adjusted by changing the SOE–receiver spacing. This fine-tunes the irradiance for the high concentration required by thermal applications, such as solar welding [21] and solar Stirling engines [22], or for better homogeneity, which is needed for concentrated photovoltaics [5]. The algorithmic optimization of the hemispherical SOE can be further modified to add more parameters or optimize other SOE designs, which saves optimization time and reduces the complexity in modeling such solar concentrator systems for various applications.

## 2. Design of the POE

Modeling the flat-spot Fresnel lens POE is based on a two-dimensional ray-tracing method, where one ray at the center of each prismatic groove is traced through the lens with the purpose of refracting that ray onto the center of the focal plane by optimizing the prism angle, $\theta$.

To simplify the design, the following assumptions were made:

1.  Solar light penetrates the Fresnel lens with no absorption, and the solar radius angle is included within the design incidence angle, $\pm \alpha$;
2.  The lens has $g$ number of grooves at an equal width of $w$, the groove order of $1 \leq i \leq g$ extends away from the lens' center;
3.  The optimization algorithm only considers the two extreme incidence angles, $\pm \alpha$, and only refracts the ray through the grooves' midpoints for the calculation of the prism angles.

The full polychromatic span of the solar spectrum was incorporated in the algorithm using the segmentation method of Yeh and Yeh [23], which was used in an earlier publication of the authors [24]. The spectrum was divided into nine wavelength intervals, each represented by a center value and a

weight factor of the total irradiant energy. Ray tracing through the POE was performed at the each of those center values, and the optimized prism dimensions were averaged out to incorporate the weight factors of all segments.

Referring to Figure 1, if a ray is incident onto the upper-surface center of the $i$th groove at an incidence half angle of $+\alpha$ with respect to the surface normal, the first refraction will happen according to Snell's law as

$$\beta = \sin^{-1}\left[\frac{\sin \alpha}{n_{lens}}\right] \tag{1}$$

where $\beta$ is the lens' top surface refraction angle and $n_{lens}$ is its refractive index. This index was calculated for each material at the center values of the segmented solar spectrum and then iterated in the optimization process. For instance, the Sellmeier equation [25] for the refractive index of poly-methyl-methacrylate (PMMA), a common material in Fresnel lens fabrication, can be used. The refracted ray hits the lower surface of the $i$th groove at an angle of $\gamma_i$ with the surface normal:

$$\gamma_i = \theta_i + \beta \tag{2}$$

Then, the second refraction takes place with an angle of $\delta_i$ found by

$$\delta_i = \sin^{-1}(\sin \gamma_i n_{lens}) \tag{3}$$

For a base thickness of $L$ and an equal groove width of $w$, the refracted ray's path length inside the $i$th groove can be computed as

$$h'_i = \left(L + \frac{w \tan \theta_i}{2}\right)\frac{\sin(90 + \gamma_i - \beta)}{\cos \gamma_i} \tag{4}$$

where $F$ is the lens' focal length. The linear vertical and horizontal distances, $A_i$ and $B_i$ respectively, that the ray travels from the $i$th groove to the focal plane are calculated by

$$A_i = F - h'_i \cos \beta + L \tag{5}$$

$$B_i = A_i \tan(\delta_i - \gamma_i + \beta) \tag{6}$$

The reference horizontal distance to the focal midpoint, $B_{i,0}$, can be found as

$$B_{i,0} = (i-1)w + \frac{w}{2} - h'_i \sin \beta \tag{7}$$

Through an iterative optimization using MATLAB®, the $i$th prism angle, $\theta_{i,+\alpha}$, can be chosen, in reference to the incidence half angle, $+\alpha$, so that the difference between $B_i$ and $B_{i,0}$ is at a minimum.

The calculations can then be repeated for the other incidence half angle of $-\alpha$, and the optimum prism angle, $\theta_{i,optimum}$, can be chosen such that

$$\theta_{i,optimum} = \frac{\theta_{i,+\alpha} + \theta_{i,-\alpha}}{2} \tag{8}$$

Each prism thickness, $h_i$, can then be calculated at the optimum, $\theta_{i,optimum}$, as

$$h_i = w \tan \theta_{i,optimum} \tag{9}$$

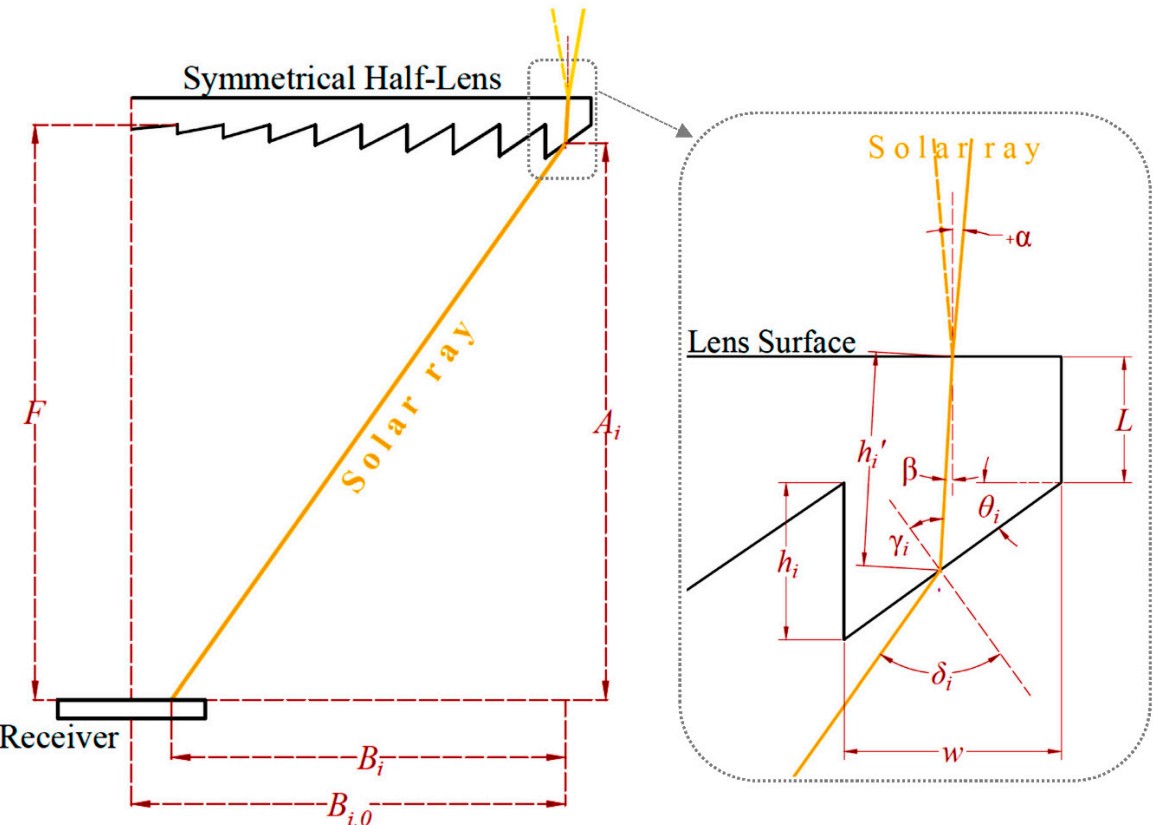

**Figure 1.** Schematic of the refraction geometry for the primary optical element (POE) design. (Half-lens shown is not to scale).

## 3. Design of the SOE

The POE optimization described in the previous section aimed to enhance the focal concentration of light, which decreases the size of the SOE needed. However, the proposed SOE optimization can also be applied to any available flat Fresnel lens, as long as the f-number, lens thickness, prism angles, and groove widths are all known.

### 3.1. Ray Tracing through SOE

With the known POE prismatic geometry, the design process for the hemispherical-domed SOE could start by solar ray tracing at the iterated SOE radius. Note that the absorption of light within the SOE was also ignored for this process. Similar to the POE optimization, the spectral segmentation of light [23] was considered while tracing rays through the SOE material, and the optimized SOE parameters were averaged according to the segments' weight factors.

In a simplified 2D consideration of the flat-spot Fresnel lens, the light incident at an angle of $+\alpha$ required two sets of equations while ray tracing, one for each side of the symmetrical lens ($L$ for left-side and $R$ for right-side). All used terms are explained in Figures 1 and 2.

While the angles of incidence and the angles of refraction at the upper POE surface, $+\alpha$ and $\beta$ respectively, are the same for both sides of the lens, the angles of incidence at the lower $i$th groove surface, $\gamma_{i,R/L}$, for the right and left sides of the lens, can be calculated in reference to Equation (2) as

$$\gamma_{i,R} = \theta_{i,optimum} + \beta \tag{10}$$

$$\gamma_{i,L} = \theta_{i,optimum} - \beta \tag{11}$$

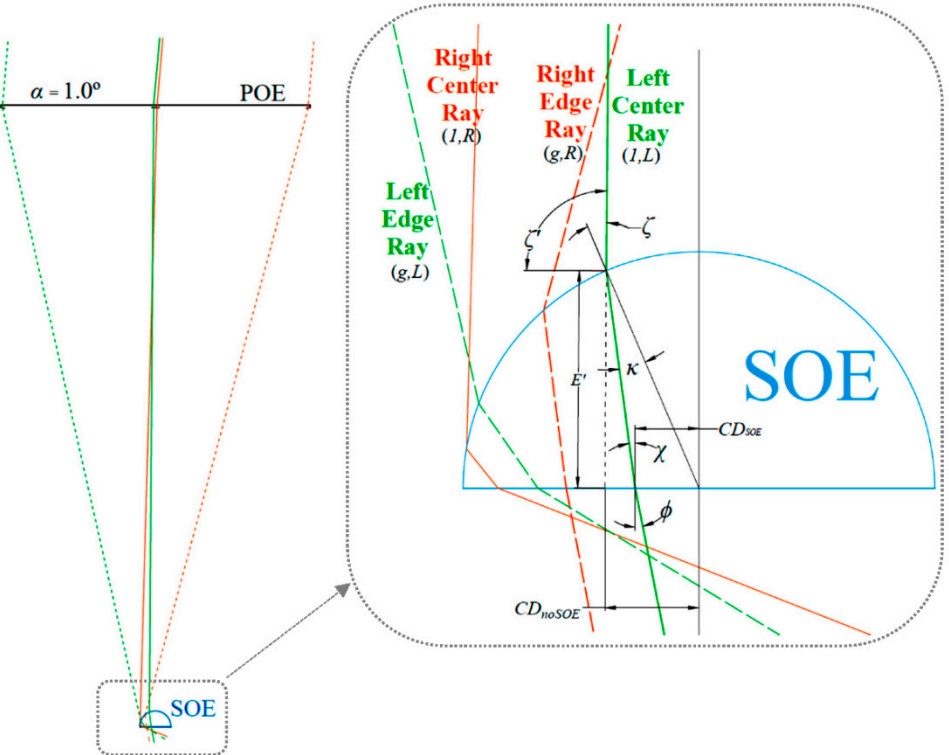

**Figure 2.** Refraction schematic through the secondary optical element (SOE). Rays from the left side of the lens are in green, and those from the right side are in orange—only four rays are shown for simplicity. (All are to scale except the incident light at the POE surface.) (The black short-dashed line is the projection of the left center ray without the SOE.)

Rays were then refracted at an angle of $\delta_{i.R/L}$ from the lower POE surface, calculated by invoking Equation (3) for both sides of the lens as

$$\delta_{i.R/L} = \sin^{-1}\left(\sin\gamma_{i,R/L}\, n_{lens}\right) \tag{12}$$

The refracted ray's path length inside the *i*th groove $h'_{i,R/L}$ was found by referring to Equation (4):

$$h'_{i,R} = \left(L + \frac{w\tan\theta_{i,optimum}}{2}\right)\frac{\sin(90 + \gamma_{i,R} - \beta)}{\cos\gamma_{i,R}} \tag{13}$$

$$h'_{i,L} = \left(L + \frac{w\tan\theta_{i,optimum}}{2}\right)\frac{\sin(90 - \gamma_{i,L} - \beta)}{\cos\gamma_{i,L}} \tag{14}$$

The angled rays making contact with the horizontal, negative *x*-axis after this second refraction, $\xi'_{i,R/L}$, can be calculated by

$$\xi'_{i,R} = 90 + (\delta_{i,R} - \gamma_{i,R} + \beta) \tag{15}$$

$$\xi'_{i,L} = 90 - (\delta_{i,L} - \gamma_{i,L} - \beta) \tag{16}$$

The linear vertical distance, $A_{i,R/L}$, for the *i*th groove on either side can be obtained using the corresponding value from Equation (5):

$$A_{i,R/L} = F - h'_{i,R/L}\cos\beta + L \tag{17}$$

Similarly, the reference linear horizontal distance, $B_{i,0,R/L}$, is given by

$$B_{i,0,R} = (i-1)w + \frac{w}{2} - h'_{i,R}\sin\beta \tag{18}$$

$$B_{i,0,L} = (i-1)w + \frac{w}{2} + h'_{i,L}\sin\beta \tag{19}$$

Without any SOE, when each refracted ray reaches the focal plane, it will be at a horizontal distance of $CD_{i,noSOE,R/L}$ from the center, which can be calculated by

$$CD_{i,noSOE,R} = A_{i,R}\tan(\delta_{i,R} - \gamma_{i,R} + \beta) - B_{i,0,R} \tag{20}$$

$$CD_{i,noSOE,L} = B_{i,0,L} - A_{i,L}\tan(\delta_{i,L} - \gamma_{i,L} - \beta) \tag{21}$$

When installing the SOE of radius $r$, each ray that hits it will have an angle $\xi_{i,R/L}$ with the normal to its curved surface, depending on the lens side it refracts from. $\xi_{i,R/L}$ is given by

$$\xi_{i,R/L} = \sin^{-1}\left[CD_{i,noSOE,R/L}\frac{\sin\xi'_{i,R/L}}{r}\right] \tag{22}$$

If the SOE has a refractive index of $n_{SOE}$, rays will get refracted into the SOE at an angle with the curved-surface normal of $\kappa_{i,R/L}$, which is calculated for both sides using

$$\kappa_{i,R/L} = \sin^{-1}\left[\frac{\sin\xi_{i,R/L}}{n_{SOE}}\right] \tag{23}$$

Each ray will then travel for a distance of $E'_{i,R/L}$ inside the SOE, which can be found as

$$E'_{i,R/L} = \frac{r\sin\left(\xi'_{i,R/L} - \xi_{i,R/L}\right)}{\sin\left(180 - \kappa_{i,R/L} - \left(\xi'_{i,R/L} - \xi_{i,R/L}\right)\right)} \tag{24}$$

Those rays will then be incident at the SOE's flat surface (dome-based surface) at an angle $\chi_{i,R/L}$ with the surface normal given by

$$\chi_{i,R/L} = 90 - \left(\xi'_{i,R/L} - \xi_{i,R/L} + \kappa_{i,R/L}\right) \tag{25}$$

Rays will also be at a horizontal distance of $CD_{i,SOE,R/L}$ from the focal center that is equal to

$$CD_{i,SOE,R/L} = \frac{r\sin\left(\kappa_{i,R/L}\right)}{\sin\left(180 - \kappa_{i,R/L} - \left(\xi'_{i,R/L} - \xi_{i,R/L}\right)\right)} \tag{26}$$

The refraction angle, $\phi_i$, off of the base of the hemispherical dome can be calculated by

$$\phi_{i,R/L} = \sin^{-1}\left(\sin\chi_{i,R/L}n_{SOE}\right) \tag{27}$$

## 3.2. SOE Geometry Optimization

With the purpose of maximizing the CAP value while minimizing both the SOE and the receiver geometry, the following three design parameters, illustrated in Figure 3a, should be optimized:

1.  The SOE radius, $r$. Minimizing this parameter will save fabrication material and cost and decrease the transmission losses through the SOE;

2.  The focus size, *FS*, which is the horizontal distance from the vertical symmetry axis to the farthest refracted ray, at the position where all refracted rays are closest to that axis. This also represents the recommended solar receiver radius. While it depends on the SOE size, the smaller this parameter is for the same acceptance angle, the higher the geometrical concentration ratio, *CG*, and the CAP values;

3.  The SOE–receiver spacing, $SRS_{opt}$, which represents the vertical distance from the *FS* to the dome's bottom surface—taking that surface at a vertical distance from the Fresnel lens equal to the focal length [5,11,15]. $SRS_{opt}$, as it also depends on the SOE size, can be further optimized by simulation for either a higher concentration of rays or a better focal irradiance homogeneity. The value obtained from the algorithm is the one achieving the highest CAP at the given SOE and *FS* values.

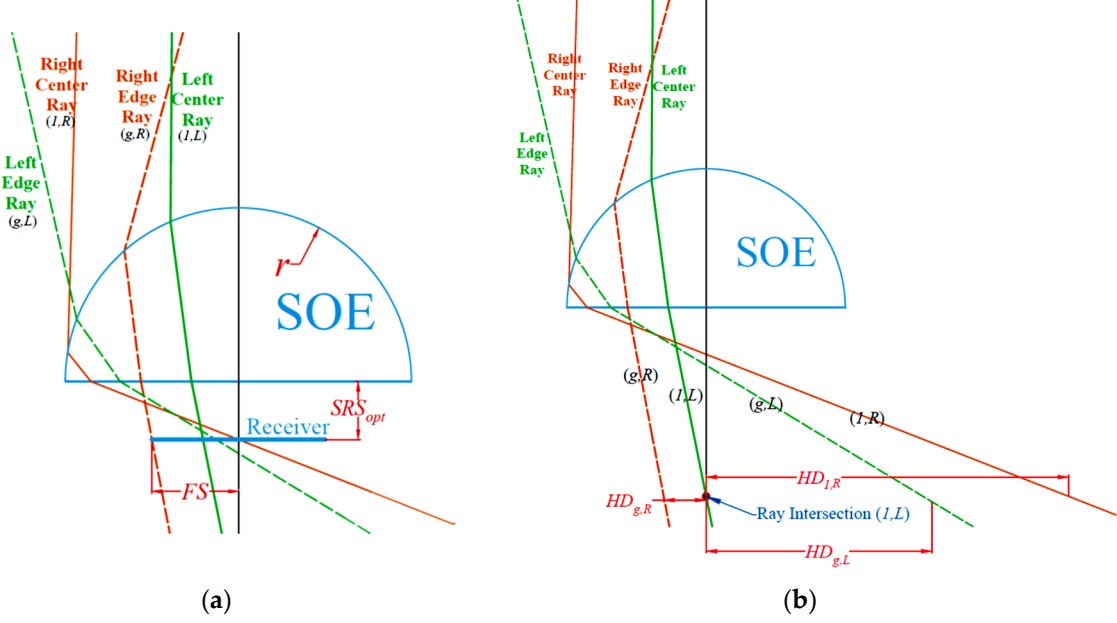

**Figure 3.** (**a**) Optimization parameters of the SOE and (**b**) ray intersection geometry explained (only the innermost and outermost rays are displayed).

The proposed optimization process accounts for rays refracted from the outermost (edge: *g*) grooves along with the innermost (1) ones and is represented by $(i, side) = [(1, R), (g, R), (1, L), (g, L)]$, taking all other rays to be refracted within those extremes. The procedure, illustrated in Figures 3b and 4, is as follows:

1.  The initial guess of the SOE radius, $r_0$, is set to the maximum value of $CD_{i,noSOE,side}$ among all four refracted rays as

$$r_0 = \max\left\{CD_{i,noSOE,side} \ : \ (i, side) = [(1, R), (g, R), (1, L), (g, L)]\right\} \tag{28}$$

The ray tracing analysis through the SOE is then carried out as per the equations of the previous sections. Noting that each *j*th iteration of the SOE radius, $r_j$, will be equally incremented by a user input value, $r_{inc}$, until convergence. The algorithm also checks for the total internal reflection (TIR) from the SOE's flat bottom and prevents it by incrementing the SOE size;

2.  The intersection point, if it exists, for each of the four rays refracted through the SOE with the vertical symmetry axis is found. The vertical distance, $SRS_{i,side}$, of each intersection point to the base of the SOE is found as

$$SRS_{i,side} = CD_{i,SOE,side} \tan\left(\left|90 - \phi_{i,side}\right|\right) \ : \ (i, side) = [(1, R), (g, R), (1, L), (g, L)] \tag{29}$$

3. For each intersection point, the corresponding horizontal distance, $HD_{i,side}$, from each of the other three rays to that intersection point is found. For example, if the intersection is for ray $(1, R)$, then the corresponding horizontal distances of the other three rays are

$$HD_{(g/1/g),\ (R/L/L)} = CD_{(g/1/g),SOE,,(R/L/L)} - \frac{SRS_{1,R}}{\tan\left(\left|90 - \phi_{(g/1/g),\ (R/L/L)}\right|\right)} \tag{30}$$

4. The maximum value of the three horizontal distances, $HD_{max}$, for each intersection point is found. This represents the horizontal distance from each intersection point to the farthest ray. For example, if the intersection is for ray $(1, R)$, then

$$HD_{max,\ intersection\ of\ (1,R)} = \max\{HD_{i,side}\ :\ (i, side) = [(g, R), (1, L), (g, L)]\} \tag{31}$$

5. The focus size, $FS$, is taken as the minimum value of all the four $HD_{max}$ values or

$$FS = \min\{HD_{max,\ intersection\ of\ (i,side)}\ :\ (i, side) = [(1, R), (g, R), (1, L), (g, L)]\} \tag{32}$$

6. The corresponding $SRS_{i,side}$ is taken as the optimum SOE–receiver spacing, $SRS_{opt}$. As a reminder, the optimum spacing here refers to the receiver position that balances the irradiance homogeneity and the concentration power. Further simulative analysis, illustrated later in Section 4.1, was used in order to discuss the fine-tuning of the SOE–receiver gap to achieve either higher concentration or better homogeneity, while keeping a relatively high CAP value.

7. The control parameter of the algorithm is $FS$. The algorithm convergence can be changed by resetting a percentile threshold, $T_s$, that tests the reduction in $FS$ for consecutive $r_j$ iterations and terminates the process when that reduction is less than the assigned threshold. Analytically, the termination happens if

$$\left|\frac{FS_j - FS_{j-1}}{FS_{j-1}} \times 100\%\right| < T_s \tag{33}$$

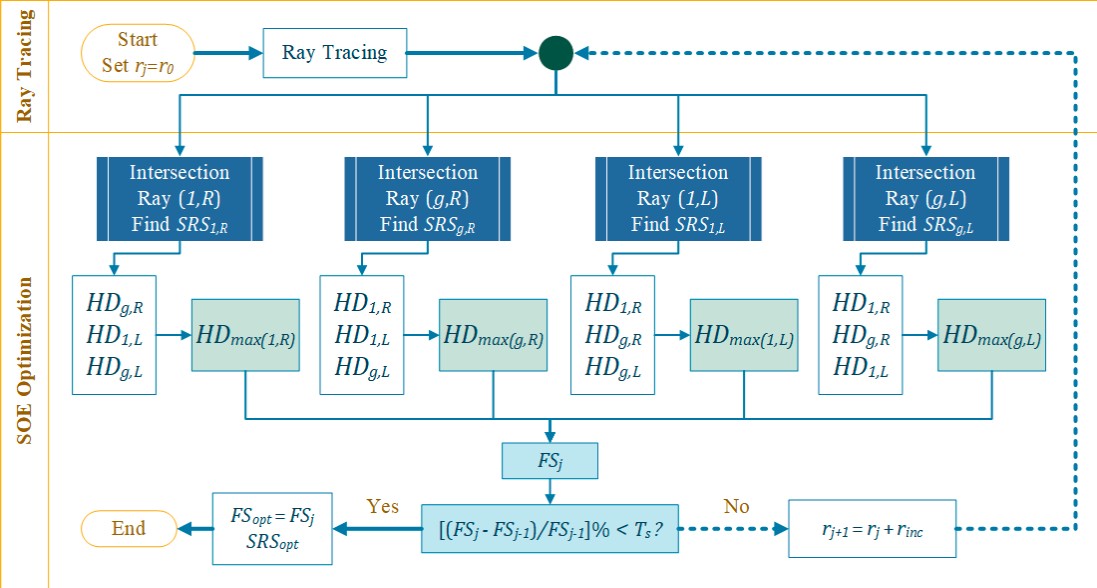

**Figure 4.** Schematic for the optimization algorithm of the SOE parameters.

An algorithmic optimization in MATLAB$^\circledR$ will iterate $r$ and optimize the resulting parameters, outputting the smallest SOE radius that can achieve the highest optical efficiency at the design incidence angle within the narrowest receiver size at a balanced distance below the SOE. The choice of the optimum SOE–receiver spacing, $SRS_{opt}$, is for the CAP value to be at a maximum.

## 4. Results and Discussion

Two case studies were selected from literature to evaluate the performance of this work's optimization. The first was for a non-optimized dome SOE, and the second was for a single-surfaced spherical SOE that was analytically optimized with ray tracing analysis.

*4.1. Case Study 1: Comparison with Non-Optimized Literature Work*

To verify the algorithmic design and optimization processes, the work of Benítez et al. [5] was selected for a case study. It compared many SOE shapes, among which a conventional non-optimized spherical dome SOE was included, based on an equal acceptance angle, $\alpha_{90\%}$, of 1.0° (the incidence angle at which the optical efficiency is 90% of the maximum value). The same 625 cm$^2$ POE was used for all. The dome SOE had a geometrical concentration, $CG$, of 257× and an f-number of 1.5, which results in a CAP value of 0.28 ($CAP = \sqrt{CG}\, \sin\alpha_{90\%}$).

For this work's POE algorithm, an equivalent area of 625 cm$^2$ and an f-number of 1.5 was used, with a design incidence angle of 1.0° and a groove width of 3 mm with PMMA lens material. The algorithm returned the prismatic characteristics in Figure 5a, indicating the prism angles, $\theta_{i,optimum}$, and groove thicknesses, $h_i$. The groove width selection was based on a sensitivity analysis that included a range of 1–5 mm values.

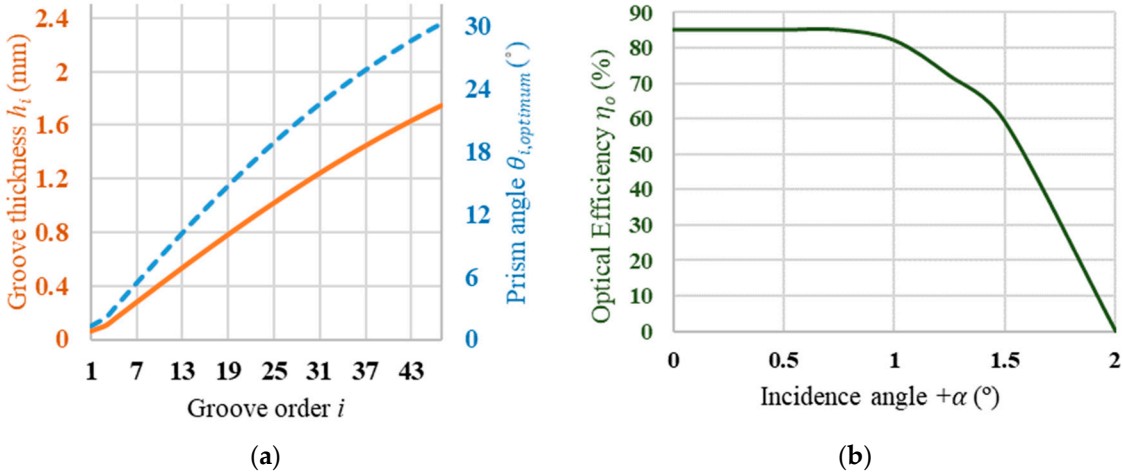

**Figure 5.** For Case 1: (**a**) prism angles (dashed blue line) and groove thickness (solid orange line) of the optimum Fresnel lens POE and (**b**) the concentrator's optical efficiency (POE + SOE) at different incidence angles.

By feeding the POE design values and applying the SOE optimization algorithm at a 1% threshold, $T_s$, with a TIR prevention scheme from the SOE base, 0.2 mm of radius increment, $r_{inc}$, and PMMA SOE material, the following concentrator parameters were obtained:

1.　Optimum SOE radius ($r$): 13.6 mm;
2.　Optimum recommended receiver radius ($FS_{opt}$): 6.65 mm;
3.　Optimum SOE–receiver spacing ($SRS_{opt}$): 5.0 mm.

Then, the concentrator was simulated with Monte-Carlo ray tracing (MCRT) via COMSOL Multiphysics$^\circledR$ version 5.3. Ray independence analysis was carried out for 10$^3$–10$^6$ rays, and a total of

$10^4$ rays were found sufficient for this simulation to save computational time. An 850 W/m$^2$ solar flux was shone on the lens, the same as that used in literature, at a range of incidence angles from 0.0° to 2.0°. The corresponding optical efficiencies, $\eta_o$, are plotted in Figure 5b, from which the acceptance angle, $\alpha_{90\%}$, was found as 1.15°. With the optimized receiver radius of 6.65 mm, the concentrator's *CG* was 449.6×, which resulted in a CAP of 0.426—about 52% higher than that by Benítez et al. [5].

The 2D ray trajectories and the focal irradiance distribution at the acceptance angle are illustrated in Figure 6a,b, respectively. Figure 6a shows all solar rays being refracted through the optimized SOE without TIR. The focal irradiance distribution, depicted by Figure 6b, peaks at 221 suns (1 sun = 1000 W/m$^2$) with poor homogeneity. Benítez et al. [5] mentioned that the irradiance homogeneity by a spherical dome SOE will be poor and can be improved by an enhanced POE design but at the expense of a reduced CAP. However, as discussed later in this section changing the SOE–receiver spacing for the same POE can also fine-tune the irradiance and produce better homogeneity.

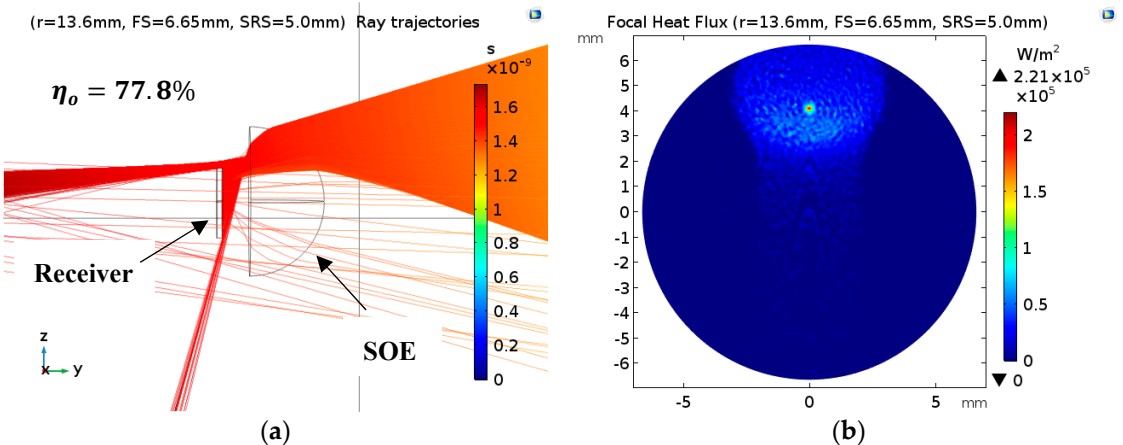

**Figure 6.** For Case 1: (**a**) ray trajectories through the optimum SOE parameters of this work at the acceptance angle of 1.15° and (**b**) the receiver's irradiance distribution in W/m$^2$ at the acceptance angle of 1.15°.

A simulative study was carried out to verify the analytical optimization. SOE parameters were iterated one at a time while fixing the other two, and the effects of changing each parameter were tested with COMSOL®. The resulting optical efficiencies are depicted against the iterated parameters in Figure 7a–c.

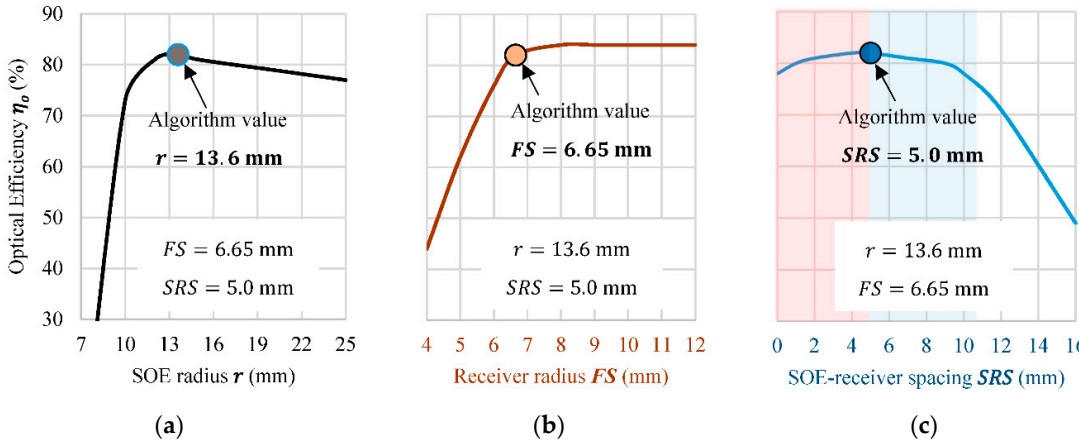

**Figure 7.** Case 1 concentrator optical efficiencies at an incidence angle of 1.0° for the iterated parameters (optimum values from the algorithm are enlarged in each curve) of (**a**) SOE radius, *r*, in mm, (**b**) receiver radius, *FS*, in mm, and (**c**) SOE–receiver spacing, *SRS*, in mm (shaded areas: left—higher concentration, right—better homogeneity).

The algorithmically optimized value of the SOE radius, *r*, returned the highest optical efficiency for a fixed receiver size and position, as seen in Figure 7a. Smaller SOEs could not catch all refracted rays from the POE, significantly reducing the efficiency. On the other hand, increasing the SOE radius maintained good efficiency but was gradually unable to effectively mitigate the focal shift, which eventually reduced the received energy again.

Widening the receiver radius, *FS*, in Figure 7b, which also means reducing the geometrical concentration ratio, *CG*, increased the optical efficiency due to the widened energy capture area. This behavior continued until converging to a maximum value. Since the actual receiver dimensions can vary, this algorithmically optimized value of the receiver radius denotes the start of convergence for the *FS*–$\eta_o$ curve at constant *r* and *SRS*.

As for the SOE–receiver spacing, *SRS*, placing the receiver too far will result in a decreased optical efficiency for a constant *CG*. It can be gleaned from Figure 7c that this distance peaks and maintains a high optical efficiency as the receiver gets closer to the SOE.

It was also noted from the simulations leading to Figure 7c that carefully increasing the value of the *SRS* (within the blue-shaded range right of the optimum) will result in a more uniform focal irradiance, while only slightly increasing the f-number of the system. On the other hand, decreasing this spacing (within the red-shaded range left of the optimum) results in a higher concentration of rays and increases the peak focal irradiance. For a better understanding, Figure 8a–f illustrates the focal irradiance distribution at 1.0° incidence for SOE–receiver spacings of 0, 5, and 12 mm.

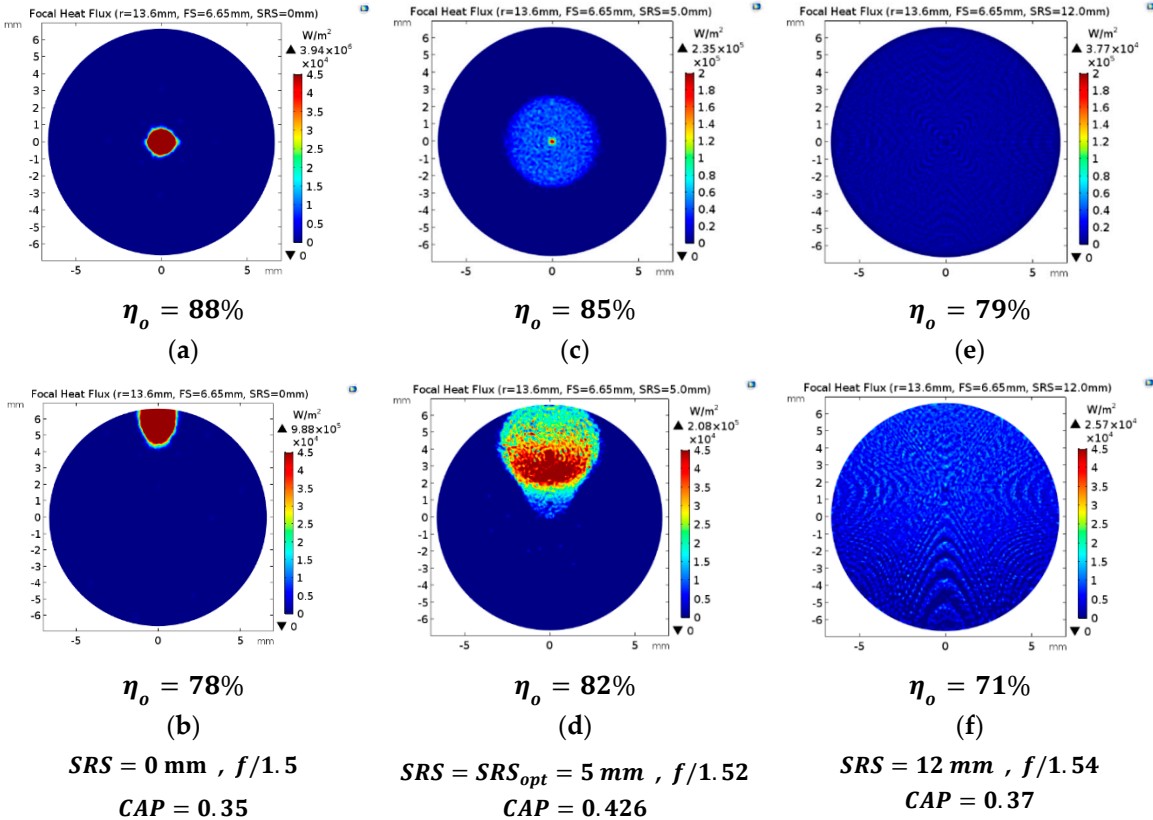

**Figure 8.** Case 1 receiver's irradiance distribution in W/m² at an incidence angle and an SOE–receiver spacing (SRS), respectively, of (**a**) normal and 0 mm, (**b**) 1.0° and 0 mm, (**c**) normal and 5 mm, (**d**) 1.0° and 5 mm, (**e**) normal and 12 mm, and (**f**) 1.0° and 12 mm. (Optical efficiencies are indicated below each figure; the f-number was taken with respect to the receiver.)

Since the purpose of the dome SOE is to increase the CAP value [5], the optimized spacing, $SRS_{opt}$, achieved the highest CAP with moderate irradiance homogeneity, as illustrated in Figure 8c,d. A highly concentrated focal flux, shown in Figure 8a,b, that peaks at 4000 suns with normal incidence can be achieved with no receiver–SOE gap at all. This is more suited for high-temperature thermal applications [21,22] but will result in a 22% CAP decrease.

A better homogeneity of the focal irradiance can be seen in Figure 8e,f, achieved by moving the receiver slightly farther from the SOE. A 12 mm gap, almost double the optimum, enhanced the uniformity of the focal flux to a plateau of about 40 suns at normal incidence and still decreased the CAP by 15%. This position is more applicable for concentrated photovoltaics.

### 4.2. Case Study 2: Comparison with Optimized Literature Work

Further verification of this work's optimization was conducted with a comparison to the work of Davis [10], in which the design of a single-surface spherical SOE was optimized. The method used by Davis [10] is different, since it did not perform the ray tracing from the top surface of the POE all the way to the solar receiver. Instead, ray tracing was conducted solely for the POE first, and its concentration ratio was found. Then, the method assumed the rays entering the SOE as collimated, and a secondary concentration ratio for the SOE was calculated. The total concentration ratio was then found using both the POE and SOE values. The design assumed no gap between the receiver and the SOE, while elongating the base of the SOE in the shape of a pillar glued to the receiver.

Davis' [10] design of interest was a 20 cm diameter Fresnel lens with an f-number of 1.37 and a geometrical concentration of 383×. MCRT simulation was conducted with $10^4$ rays at a refractive index of 1.5. The system used a domed-pillar SOE with a 9.8 mm radius domed top, a pillar hight of 8 mm, and a solar receiver glued to the bottom. At a 1.0° design acceptance angle, the concentrator had a CAP value of 0.34.

For this case study comparison, an equally sized PMMA-Fresnel lens POE was modeled at the same f-number and design incidence angle as Davis' [10]. The grooves had an equal width of 5 mm, and its resulting prism angles were fed into the SOE algorithm and yielded the following optimized PMMA–SOE parameters:

1.  Optimum SOE radius ($r$): 11.4 mm;
2.  Optimum recommended receiver radius ($FS_{opt}$): 4.83 mm;
3.  Optimum SOE–receiver spacing ($SRS_{opt}$): 1.0 mm.

The same criteria of Case 1 was employed at a 1% threshold, $T_s$, a TIR prevention scheme, and a 0.2 mm radius increment, $r_{inc}$. It can be noted that, even though the optimized SOE radius of this work is about 16% larger than that used by literature, the overall SOE volume decreased by 29.4%. This is due to the removal of the elongated cylindrical portion of the SOE used by Davis' [10] design.

An MCRT simulation was conducted for this work's optimized POE–SOE design with a flux of 1000 W/m$^2$ and $10^5$ rays. The receiver radius was increased in the simulation from the optimized value of 4.83 to 5.11 mm, in order to maintain the concentration ratio used by Davis [10]. An acceptance angle of 1.31° was attained, as inferred from Figure 9, which boosted the CAP by 32.4% to 0.45—in comparison to the literature value.

Smooth and TIR-free ray trajectories at the acceptance angle of 1.31° are illustrated in Figure 10a. Since this work's optimization aims to maximize the CAP, poor focal irradiance homogeniety is notable in Figure 10b, with the flux peaking at about 1 MW/m$^2$. Better focal homogeniety can be achieved by further fine-tuning of the SOE–receiver gap.

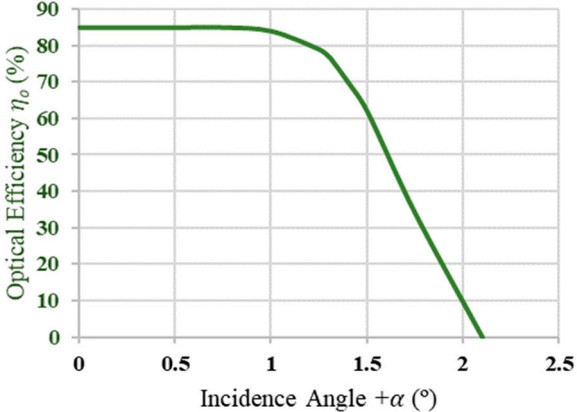

**Figure 9.** The concentrator's optical efficiency (POE + SOE) at different incidence angles for Case 2.

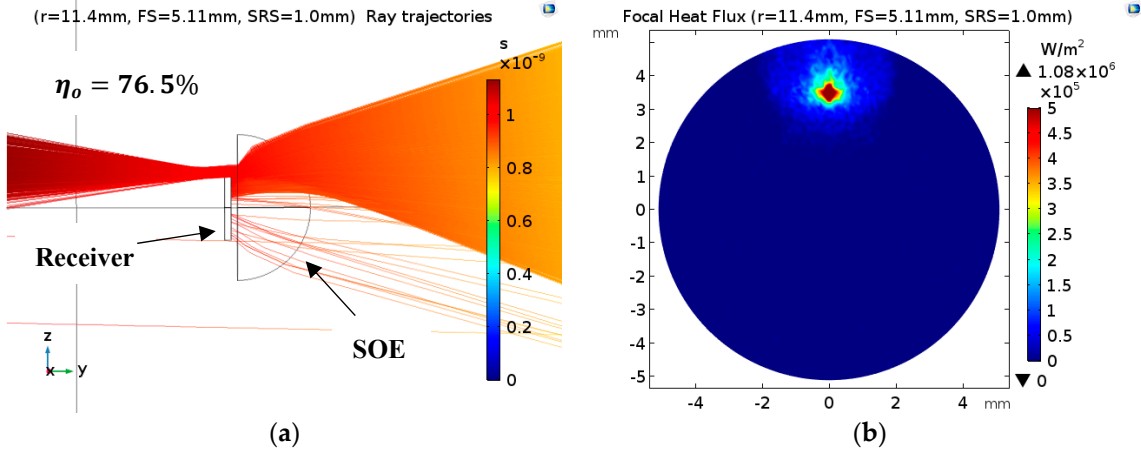

**Figure 10.** For Case 2: (**a**) ray trajectories through the optimum SOE parameters of this work at the acceptance angle of 1.31° and (**b**) the receiver's irradiance distribution in W/m² at the acceptance angle of 1.31°.

## 5. Conclusions

This work explained the optimization analysis for the Fresnel lens POE and the dome-shaped SOE, which can significantly enhance the concentration acceptance product, CAP. The importance of this work lies in the deterministic and comprehensive optimization of the concentrator, utilizing polychromatic ray tracing, trigonometry, and spherical geometry in a MATLAB® algorithm. The optimum design outputs the POE prism geometry, the SOE radius, the receiver–SOE spacing, and the recommended receiver radius. The designed concentrator can then be simulated with COMSOL® to evaluate the optical efficiency, focal irradiance distribution, and ray trajectories.

A proof of concept case study was selected from literature for a 625 cm² spot-lens POE with an f-number of 1.5 and a spherical dome SOE. An optimized 13.6 mm SOE radius, 84% smaller in volume than that used by the referenced work, increased the concentrator's acceptance angle from 1.0° to 1.15°. A 74.9% increase in the geometrical concentration was imposed by the optimized receiver radius, which, along with the increased acceptance angle, enhanced the CAP value by 52%, compared to literature.

The coherence of the optimization results was confirmed with a simulated comparative analysis, which iterated the SOE parameters up to 200% of the optimum values. The optimized parameters returned the peak efficiency in each case. The focal irradiance homogeneity was analyzed for different SOE–receiver gap sizes. A no-gap design increased the concentration power, while almost doubling the gap size significantly enhanced the homogeneity. However, the CAP value was decreased by 20% and 15% respectively, compared to the algorithmically optimized SOE–receiver spacing.

A second case study of an optimized dome-shaped SOE from literature was also compared to this work. The POE was a 314.2 cm$^2$ spot-lens with a 383× geometrical concentration and a 27.4 cm focal length. This work's algorithm yielded an optimized dome SOE with a radius of 5.7 mm, which was about 30% smaller in volume compared to that optimized by literature. The CAP value was boosted from 0.34 to 0.45 due to the 31% increase in the acceptance angle. A comparative summary of both case studies is listed in Table 1.

**Table 1.** Results comparison with the literature's case studies.

| Parameter | Case Study 1 | | Case Study 2 | |
|---|---|---|---|---|
| | Benítez et al. [5] | This Work | Davis [10] | This Work |
| POE size (cm$^2$) | 625 | 625 | 314.2 | 314.2 |
| f-number | 1.5 | 1.517 | 1.37 | 1.37 |
| Geometrical concentration | 257 | 449.6 | 383 | 383 |
| Dome SOE radius (mm) | 25 | 13.6 | 9.81 | 11.4 |
| Total SOE volume (cm$^3$) | 32.72 | 5.26 | 4.40 | 3.10 |
| SOE–receiver spacing (mm) | No spacing | 5.0 | No spacing | 1.0 |
| Acceptance angle (°) | 1.0 | 1.15 | 1.0 | 1.31 |
| *CAP* | 0.28 | 0.426 | 0.34 | 0.45 |

This work's method can drive further research on the algorithmic optimization of different SOE shapes and can be enhanced to include more parameters, such as optimizing the SOE material and the number of POE grooves.

On the other hand, while this optimization is especially designed to maximize the CAP value, future research will work on adjusting the algorithmic approach and the analytical parameters to achieve a uniform focal irradiance for the application of concentrated photovoltaics. This will be advantageous due to the small and simple SOE design, aiding a better area utilization, easier and cheaper manufacturing, and a higher system efficiency.

An ongoing research of the authors is also focused on prototyping the Fresnel lens POE with micro-machining and hot embossing and the dome SOE with glass molding. Then the prototype system will be tested with both thermal applications, such as the solar Stirling engine and solar welding, and a CPV application.

**Author Contributions:** Conceptualization, methodology, software, validation, formal analysis, investigation, resources, data curation, and writing—original draft preparation: H.Q.; writing—review and editing, visualization, supervision, and funding acquisition: S.W. and W.Z.

**Funding:** This work was supported by the University of North Texas College of Engineering, Department of Mechanical and Energy Engineering, through the faculty's start-up funding.

**Acknowledgments:** The authors would like to acknowledge the support offered by the College of Engineering at the University of North Texas.

**Conflicts of Interest:** The authors declare no conflict of interest.

## Nomenclature

| | |
|---|---|
| *A* | Ray's vertical travel distance from groove to focal plane [mm] |
| *B* | Ray's horizontal travel distance from groove to focal plane [mm] |
| $B_0$ | Reference horizontal distance from groove to focal plane midpoint [mm] |
| *CG* | Geometric concentration ratio |
| $CD_{noSOE}$ | Horizontal distance from refracted ray to symmetry axis at focal plane (no SOE) [mm] |
| $CD_{SOE}$ | Horizontal distance from refracted ray to symmetry axis at focal plane (with SOE) [mm] |
| *E′* | Ray's path length inside the SOE [mm] |
| *F* | Focal length [mm] |
| *FS* | Recommended solar receiver radius [mm] |

| | |
|---|---|
| *g* | Number of grooves |
| *h* | Groove thickness [mm] |
| *h′* | Ray's path length inside the lens [mm] |
| *HD* | Horizontal ray distance to intersection point [mm] |
| *i* | Groove order |
| *L* | Lens base thickness [mm] |
| *n* | Refractive index |
| *r* | SOE radius [mm] |
| *SRS* | SOE–receiver spacing [mm] |
| $T_s$ | Optimization threshold [%] |
| *w* | Groove's equal width [mm] |
| *Greek symbols* | |
| $\alpha$ | POE's top surface incidence angle [°] |
| $\alpha_{90\%}$ | Acceptance angle [°] |
| $\beta$ | POE's top surface refraction angle [°] |
| $\gamma$ | POE's bottom surface incidence angle [°] |
| $\delta$ | POE's bottom surface refraction angle [°] |
| $\eta_o$ | Lens optical efficiency [%] |
| $\theta$ | Prism inclination angle [°] |
| $\kappa$ | SOE's curved surface refraction angle [°] |
| $\xi'$ | Angle with horizontal after second refraction from POE [°] |
| $\xi$ | SOE's curved surface incidence angle [°] |
| $\phi$ | SOE's flat surface refraction angle [°] |
| $\chi$ | SOE's flat surface incidence angle [°] |

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
