# Peer review of "Algorithmically Optimized Hemispherical Dome as a Secondary Optical Element for the Fresnel Lens Solar Concentrator"

_applsci, doi:10.3390/app9132757_

Round 1

Reviewer 1 Report

In this study, an algorithm that couples the optimization of a Fresnel-lens POE with a dome-shaped SOE is presented. The optimized concentrator is compared with the work of Benitez et. al [5]. A MCRT analysis with COMSOL is done in order to quantify the optical behaviour of the system.

I have some questions:

1)           The authors states that laws of polychromatic light refraction along with trigonometry and spherical geometry have been utilized to optimize the POE grooves, SOE radius, receiver size, and SOE-receiver spacing. But in Sections 2 and 3 I didn't see the range of wavelenght values used. Is is mandatory to add in the manuscript the spectrum of the source used for this "polychromatic ray tracing" mentioned in line 65. Otherwise, this algorithm is not a “polycromatic optimization”.

2)           The authors used the work of Benítez et al. [5] for comparing its own algorithm. The problem here, is that the purpose of the study presented by Benitez et al. was to present a LPI’s Fresnel Kohler (FK) concentrator optimized and compare it with other five more conventional CPV concentrators. One of those conventional CPV had a dome-shaped SOE and is the one used to compare with the own optimized by the authors. Hence, the authors don't use a solar concentrator optimized in order to compare and validate  its own algorithm for achiving higher geometrical concentration ratio ?? and the ??? values. This is the reason why the CAP value of the proposed algorithm is greater than the no optimized geometry presented in [5]. In my opinion, and in order to demonstrate that the optimized algorithm presented in this manuscript has novelty, it is necessary find other cases in the literature with dome-shaped SOE but optimized. If it is not possible, the only comparison with the spherical dome of [5] lacks of interest because in [5] it was not optimized.

3)           The authors states that “It can also be noted that many literature did not account for a solar incidence angle in the design phase of the SOE, while others did not optimize the POE along with the SOE to further fine-tune the SOE geometry and enhance the optical performance. Another issue was the lack of the polychromatic representation of light in the SOE design phase, which can deviate the simulated analysis from the experimental results.” It is mandatory to add references of journals with high impact factor in order to justify all of the lack in methods mentioned in this paragraph.

4)           An extensive editing of English language and style is required of this manuscript.

Author Response

Please see the attachment.  All revised parts are highlighted in the manuscript.

Reviewer 2 Report

The manuscript is very well structured and its results are clearly presented. 

However, I would suggest to introduce a clearer and more extensive analysis regarding the applications of the obtained results for concentrated photovoltaics.

Author Response

(The authors gave the same response as above.)

Round 2

Reviewer 1 Report

My congratulations to the authors. They have considered all my suggestions and now I really believe that the paper has achieved important improvements. In my opinion the manuscript is ready to be published in the Journal.